# Validity of the CALERA Research Sensor to Assess Body Core Temperature during Maximum Exercise in Patients with Heart Failure

**DOI:** 10.3390/s24030807

**Published:** 2024-01-26

**Authors:** Antonia Kaltsatou, Maria Anifanti, Andreas D. Flouris, Georgia Xiromerisiou, Evangelia Kouidi

**Affiliations:** 1FAME Laboratory, Department of Physical Education and Sport Science, University of Thessaly, 42100 Trikala, Greece; akaltsat@gmail.com (A.K.); andreasflouris@gmail.com (A.D.F.); 2Sportsmedicine Laboratory, Department of Physical Education and Sport Science, Aristotle University of Thessaloniki, 57000 Thermi, Greece; manyfant@phed.auth.gr; 3Department of Neurology, University Hospital of Larissa, University of Thessaly, 41110 Larissa, Greece; geoksirom@med.uth.gr

**Keywords:** wearable sensors, body core temperature, chronic heart failure, thermoregulation

## Abstract

(1) Background: It is important to monitor the body core temperature (Tc) of individuals with chronic heart failure (CHF) during rest or exercise, as they are susceptible to complications. Gastrointestinal capsules are a robust indicator of the Tc at rest and during exercise. A practical and non-invasive sensor called CALERA Research was recently introduced, promising accuracy, sensitivity, continuous real-time analysis, repeatability, and reproducibility. This study aimed to assess the validity of the CALERA Research sensor when monitoring patients with CHF during periods of rest, throughout brief cardiopulmonary exercise testing, and during their subsequent recovery. (2) Methods: Twelve male CHF patients volunteered to participate in a 70-min protocol in a laboratory at 28 °C and 39% relative humidity. After remaining calm for 20 min, they underwent a symptom-limited stress test combined with ergospirometry on a treadmill, followed by 40 min of seated recovery. The Tc was continuously monitored by both Tc devices. (3) Results: The Tc values from the CALERA Research sensor and the gastrointestinal sensor showed no associations at rest (r = 0.056, *p* = 0.154) and during exercise (r = −0.015, *p* = 0.829) and a weak association during recovery (r = 0.292, *p* < 0.001). The Cohen’s effect size of the differences between the two Tc assessment methods for rest, exercise, and recovery was 1.04 (large), 0.18 (none), and 0.45 (small), respectively. The 95% limit of agreement for the CALERA Research sensor was −0.057 ± 1.03 °C. (4) Conclusions: The CALERA sensor is a practical and, potentially, promising device, but it does not provide an accurate Tc estimation in CHF patients at rest, during brief exercise testing, and during recovery.

## 1. Introduction

Global warming and the resulting climate change present a major challenge to public health [1]. One of the adverse effects of climate change is the increase in the occurrence and severity of heat waves due to rising temperatures [2]. As heatwaves become more frequent, there is a growing concern about the impact on vulnerable individuals, particularly those with pre-existing heart conditions, such as chronic heart failure (CHF) [3,4]. According to a recent multinational study conducted across twenty-seven countries, exposure to extreme hot and cold temperatures was found to be associated with an increased risk of mortality from various common cardiovascular conditions [5]. Indeed, individuals with CHF are particularly susceptible to the detrimental effects of heat exposure [3], as their insufficient heart function has to deal with an additional strain, which can worsen their symptoms and potentially lead to cardiovascular events [6]. Accordingly, a recent meta-analysis highlighted the importance of developing and implementing preventive measures to decrease the incidence of cardiovascular events during hot periods [6].

Individuals with CHF may experience specific abnormalities that can worsen their health and increase the risk of cardiovascular events during heat exposure. Heat exposure can cause excessive sweating, leading to fluid loss and dehydration [7]. Medications commonly prescribed for CHF, like diuretics, can further increase the risk of dehydration and electrolyte imbalances, which can be augmented in the presence of heat exposure, potentially leading to complications such as dizziness, weakness, or cardiac arrhythmias [8]. In addition, individuals suffering from CHF may experience difficulties regulating their core body temperature in response to heat due to impaired thermoregulatory mechanisms, and this is further augmented when they engage in physical activity and especially maximal exercise [9]. This impaired thermoregulation raises the risk of heat-related illnesses, including heat exhaustion or heatstroke. Regular physical activity and exercise training are strongly recommended for patients with CHF [10]. However, outdoor workouts in high temperatures and humidity may put them at risk of heat exhaustion or heat-related adverse events [11], which makes it crucial to develop strategies to protect them. In this regard, wearable core body temperature monitoring technology can play a critical role in monitoring and protecting individuals at risk.

Recently, new technology has been developed that achieves the measurement, data logging, and transmission of temperature at the gastrointestinal tract to be used as an indicator of body core temperature (Tc) [12]. The technology involves ingestible capsules that measure, store, and transmit temperature data wirelessly to a dedicated receiver device. This method remains invasive, yet it is more comfortable and practical compared to methods such as esophageal, bladder, and rectal temperature measurements [13,14]. By using ingestible gastrointestinal telemetric capsules, healthcare professionals can accurately and continuously monitor the Tc without causing discomfort associated with other methods. However, assessing the Tc using ingestible telemetric capsules remains an invasive and highly expensive method, with costs surpassing EUR 70 per participant.

The availability of a non-invasive sensor that can monitor the Tc at rest but also during exercise or other forms of physical work would offer numerous benefits in clinical settings. Such a sensor would assist with heat acclimation training, prevent heatstroke, and provide important diagnostic information in clinical settings. For patients with CHF, who face a high risk for heat illness or must work in hot conditions, having access to a non-invasive sensor that can alert them to changes in their Tc is particularly important. The CALERA Research sensor, a product of greenTEG A.G. based in Rümlang, Switzerland, is an example of such technology and is now available on the commercial market [15]. The CALERA Research sensor was built on the technology that powered its prototype, the CORE sensor. However, according to the manufacturer, the CALERA Research sensor offers features specifically tailored to scientific studies, going beyond the capabilities of the CORE sensor. This product assesses skin temperature, heat flux, and heart rate and employs machine learning algorithms to determine the Tc. There is only one study to date that has examined its use in healthy individuals after COVID-19 vaccination (12). Consequently, this investigation aims to compare the validity and accuracy of the CALERA Research sensor with Tc measurements during rest, exercise testing, and recovery in patients with CHF.

## 2. Materials and Methods

### 2.1. Participants

Social media advertisements were used to recruit study participants. Inclusion criteria were as follows: age between 19 and 65 years (Table 1); patients with CHF diagnosis and NYHA class ≤ III; stable clinical condition for at least six months; and abstinence from any form of regular exercise in the past year. Exclusion criteria were as follows: unstable angina, recent myocardial infarction, uncontrolled hypertension, chronic obstructive pulmonary disease, insulin-dependent diabetes mellitus, and severe neurological or orthopedic problems that would hinder their participation in the exercise program.

Before participating in this study, all volunteers were fully informed about this study’s purpose and procedures, and they provided written informed consent in accordance with the Ethical Committee of Aristotle University. This study was conducted according to the principles of the Declaration of Helsinki and was approved by the Ethics Committee of the Aristotle University of Thessaloniki (protocol no.: 159/2023).

### 2.2. Study Design

The objective of this study was to evaluate the reliability of the CALERA Research sensor at rest, during exercise stress testing, and through the recovery phase (as depicted in Figure 1). Participants were required to visit the laboratory on two distinct occasions for these assessments. The evaluation process comprised both physical examinations and exercise stress tests. All tests were scheduled between 09:00 and 11:00 in the morning. Participants were instructed to consume only a light breakfast about two hours prior to the examination and were advised against smoking, consuming coffee, or drinking alcohol for a minimum of 12 h leading up to the measurements.

Volunteers participated in a preliminary screening session. During the preliminary session, they provided a brief overview of this study’s objectives and procedures and prospective participants. To ensure the experimental results’ accuracy and the participants’ comfort, they were introduced to all the equipment and measurements they would encounter. Then, a multidimensional assessment, which included demographic and clinical parameters, was conducted to ensure their eligibility to participate in this study. This ensured that participants met this study’s inclusion criteria. Anthropomorphic data were then collected. Body height and mass were determined via a stadiometer (Seca 206, Hamburg, Germany) and a digital weighing terminal (Seca 877, Hamburg, Germany), respectively, and used to calculate body mass index. In order to maintain consistency, the anthropometric data reported are those that were measured at the beginning of the experimental trial.

The second visit was the experimental visit. Participants were asked to swallow an ingestible temperature capsule to record the visceral temperature, used as an indicator of the Tc. The time between swallowing the capsule and the onset of measurements was 121 ± 17 min. Previous research has shown that the time following capsule ingestion does not influence the validity of Tc measurements during exercise [16]. Additionally, each participant was equipped with a heart rate chest strap with an attached CALERA Research sensor. The total duration of the protocol was 70 min in a laboratory environment controlled at 28 °C air temperature and 39% relative humidity. After remaining calm for 20 min in a seated position, participants underwent a symptom-limited stress test using the Bruce protocol combined with ergospirometry. The recovery phase lasted for 40 min. Participants were instructed to sit calmly in a chair while connected to an electrocardiogram (ECG). They waited until their heart rate reached normal levels before removing the equipment. This 40-min recovery phase allowed us to monitor the return to baseline levels for several physiological parameters. Once their heart rate normalized, they were assisted in safely removing all equipment.

### 2.3. Measurements

Upon each arrival for each experimental trial, participants’ body mass was measured, and participants provided a urine sample to confirm euhydration (urine specific gravity: ≤1.025) and donned a t-shirt, shorts, socks, and athletic shoes.

#### 2.3.1. Telemetric Intestinal Temperature Device

The Tc was recorded continuously using a telemetric capsule and receiver (e-Celsius Performance^®^, BodyCap, Caen, France). The capsule is compact, measuring 17.6 mm in length and 8.6 cm in diameter and weighing 1.2 g. It is connected to a Bluetooth monitor and has a maximum sampling time of 30 s.

#### 2.3.2. Tc with Calera Research

The Tc was also measured using the CALERA Research sensor (greenTEG A.G., Rümlang, Switzerland) which was attached to a heart rate chest strap. According to the manufacturer’s instructions, the sensor was positioned on the torso/chest approximately 20 cm below the armpit using a heart rate monitor strap. Based on the manufacturer’s manual, the sensor was paired with a heart rate monitor during exercise for increased accuracy. The sensor used for the research was the CALERA Research version, which collects data every second and has a storage capacity of 6 days. According to the manufacturer, the accuracy of the Tc data provided by the device is 0.11 ± 0.34 °C [17]. The data stored on the device were retrieved by using the CALERA Research app on an Android device and later transferred to a computer for further analysis.

#### 2.3.3. Cardiopulmonary Exercise Testing

As part of this study, the participants underwent a cardiopulmonary test on a Trackmaster treadmill (Full Vision Inc., Newton, KS, USA) using a Bruce protocol until they showed symptoms. The Bruce protocol is a diagnostic tool to estimate both cardiac functionality and an individual’s physical fitness. This test requires the patient to walk on a treadmill, with electrodes affixed to their chest, to monitor cardiac activity. Every three minutes, the treadmill’s speed and incline increase and consist of seven progressive stages. The Bruce treadmill test calculates the participant’s maximum oxygen uptake through a specialized formula, tracking their endurance as the intensity augments.

The test was closely monitored, with the ECG of each patient being continuously monitored and recorded at the end of each stage, in combination with blood pressure. Expired gases were analyzed using the MedGraphics Breeze Suite CPX Ultimaergospirometer device (Medical Graphics Corp, St Paul, MN, USA), which underwent standard calibration before each test. The measurements taken during peak exercise included systolic and diastolic blood pressure, heart rate, exercise time, ventilator anaerobic threshold, pulmonary ventilation, VO_2_–heart rate relationship, and the slope of expired minute ventilation for carbon dioxide output and the slope of expired minute ventilation for carbon dioxide output. The peak oxygen consumption (VO_2_peak) was determined as the highest VO_2_ obtained, characterized by a plateau in oxygen uptake despite further increases in work rate. Obtained VO_2_peak values were considered maximal when the respiratory exchange ratio was greater than 1.10. The software defines the VO_2_ above, in which VCO_2_ increases faster than VO_2_ without hyperventilation. Overall, the test results provide valuable insight into the cardiopulmonary health of the participants.

#### 2.3.4. Sample Size Estimation

The minimum required sample size was determined based on a desired maximum difference of 0.3 ± 0.25 °C, which is considered adequate from a thermophysiological and clinical point of view. For a non-invasive device, to be able to provide Tc estimates within 0.3 °C from gastrointestinal capsules is certainly considered adequate and would not change the thermophysiological and/or clinical outcome/recommendations from any relevant testing. To calculate the minimum required sample size, the G*Power 3.0 software was employed with the “Means: Difference between two dependent means (matched pairs)” method. Statistical power (1 − β) was set to 0.95, aiming for a 95% chance of detecting a significant effect if it truly exists in the population. Additionally, an α error probability (significance level) of 0.05 was used, a standard choice for hypothesis testing. The calculation of the minimum required sample size used a two-sided test relying on Cohen’s effect size dz. The resulting minimum required sample size was 12 participants.

### 2.4. Statistical Analyses

We analyzed data normality using the Kolmogorov–Smirnov and Shapiro–Wilk tests and expressed values as means ± standard deviations (SDs). We used a paired *t*-test to evaluate the differences between the Tc data from the CALERA Research sensor and those of the gastrointestinal temperature sensor. We used Pearson’s correlation to measure the relationship between CALERA Research and gastrointestinal Tc measurements. We considered a difference of ≤0.3 °C between the devices to be acceptable. We calculated the effect size using Cohen’s d to quantify the magnitude of the difference between the Tc measurements obtained by the CALERA Research device and the e-Celsius^®^ system. Cohen’s d is a standardized measure of effect size that helps to determine the practical significance of the observed differences. Moreover, Bland–Altman analysis was used to assess the agreement between the two measurements [18]. This technique calculates the mean difference between both devices and the limits of agreement of this mean (95% confidence interval) [18]. Statistical significance was set at *p* < 0.05.

## 3. Results

Fourteen enrolled patients from a single center agreed to participate in this study, but two patients had to be excluded due to CALERA Research sensor failure to record their Tc due to technical problems (likely due to excessive sweating). Accordingly, the results of the 12 CHF patients were analyzed and presented. The clinical characteristics of the participants are presented in Table 1.

The maximum exercise time was 9.3 ± 0.6 min, and the VO_2_peak was 23.2 ± 5.45 mL/min/kg. Figure 2 presents the mean Tc of the participants, measured with the gastrointestinal sensor and the CALERA Research sensor. The analysis was further performed separately for rest, exercise, and recovery periods and showed that the mean temperature difference was statistically significant for the rest period (*p* < 0.001), exercise periods (*p* = 0.004), and recovery (*p* = 0.008) (Table 2). The Tc values obtained from the CALERA Research sensor and the gastrointestinal sensor showed no associations at rest (r = −0.059, *p* < 0.001) and during exercise (r = −0.015, *p* = 0.829), as well as a weak association during post-exercise recovery (r = 0.292, *p* < 0.001) (Table 2). The effect size (Cohen’s d) of the differences between the two Tc assessment methods for rest, exercise, and recovery was 0.40, 0.38, and 0.28 (all small), respectively.

A Bland–Altman plot is presented in Figure 3. Throughout the entire trial, the 95% limit of agreement between the Tc data from the gastrointestinal pill and the CALERA Research sensor was −0.043 ± 0.72 °C.

## 4. Discussion

Our study was designed with the primary objective of evaluating the validity of the recently developed CALERA Research device in measuring Tc in patients with CHF during rest, exercise testing, and recovery periods. The assessment involved a comparison of the results obtained by the CALERA sensor with those obtained by the e-Celsius^®^ gastrointestinal capsule, which is recognized as a robust indicator of the Tc. Our findings suggest no or weak correlations between the measurements obtained by the CALERA Research device and those of the gastrointestinal capsule during rest, exercise, and recovery. The core temperature values obtained by the two methods were statistically significantly different at rest, exercise, and recovery, while the limits of agreement were relatively wide (mean difference of −0.04 ± 0.72 °C). Τhe comparative difference between the two methods (gastrointestinal capsule and CALERA sensor) is around 0.3 °C across all stages of the experiment. The results imply that while the CALERA sensor offers a convenient method for estimating core temperature, its accuracy in providing precise core temperature readings in CHF patients, both at rest and during exercise testing, might be limited.

There is only one existing study that has examined the accuracy and consistency of the CORE sensor during exercise [19]. This study involved 12 men who underwent two different heat load protocols. The first protocol consisted of two identical 60-min sessions of steady-state cycling in a laboratory with low-to-moderate heat load, while the second protocol involved 13 men undergoing moderate-to-high heat load by cycling for 90 min in a laboratory with elevated humidity levels. These protocols aimed to evaluate the performance of the CORE sensor under varying heat stress conditions. Despite the heat load levels, the CORE sensor’s body temperature readings did not match well with the rectal temperature. In fact, around 50% of the paired measurements showed differences greater than the predefined threshold of 0.3 °C. Based on these results, the authors concluded that the claim by the CORE sensor manufacturer is not substantiated.

A more recent study evaluated the precision of the CORE sensor in monitoring the Tc of 30 acute stroke patients admitted to a dedicated stroke unit [17]. This study involved a comparison between the CORE sensor and an infrared tympanic temperature thermometer. The study reported adequate agreement (mean difference of 0.11 ± 0.34 °C) between the Tc estimates provided by the CORE sensor and the temperature measurements obtained from the tympanic thermometer. Based on the manufacturer, the CALERA Research sensor has functionalities and services tailored for scientific studies. A recent study analyzed the performance of CALERA Research in 33 healthy individuals who had received COVID-19 vaccinations [15]. The results showed that the limits of agreement were between −0.67 °C and +0.93 °C, which is comparable to the bias and limits of agreement found in commonly used tympanic membrane thermometers. According to the Bland–Altman statistics, the CALERA Research device worn on the wrist predicted an average bias of 0.11 °C for core body temperature when compared to the reference method for measuring the Tc [15].

In individuals, heat dissipation and thermal regulation are primarily achieved through skin blood flow and sweating, accompanied by cardiovascular adjustments under autonomic control [20,21]. Impairments in these mechanisms can compromise thermal regulation during exercise or exposure to elevated temperatures. Heat intolerance may be exacerbated in CHF patients, who often have impaired cardiovascular and autonomic function [3]. Real-time monitoring of daily Tc is promising for several reasons. By monitoring the Tc in real time, healthcare providers can accurately assess a patient’s hemodynamic status and intervene promptly when necessary. The Tc plays a critical role in regulating cardiac output and oxygen demand. An elevated Tc can increase metabolic demands, leading to an increased heart rate and cardiac workload. In individuals with CHF, excessive demand can worsen their condition. Healthcare providers can detect changes and adjust treatment strategies to optimize cardiac output and oxygen demand by monitoring the Tc. In this study, it was observed that participants’ core temperatures did not exceed 38 °C, typical for individuals working in hot environments [22]. Given that our participants were CHF patients, we performed our measurements in a controlled laboratory setting, maintaining an air temperature of 28 °C and a relative humidity of 39%. The heat exposure of this environment is markedly lower than that frequently reported in occupational settings or other environments linked with elevated core body temperatures. Nevertheless, it is important to note that the core body temperature of our CHF patients, as recorded by the gastrointestinal tract capsule, did not return to baseline levels within the 40 min recovery period. This may be caused by medication related to CHF and it is important to know when prescribing repeated exercise bouts in CHF patients, especially since exercise training is recommended for patients with CHF and it is included in cardiac rehabilitation programs.

Despite the valuable insights gained from our study, several limitations should be acknowledged. Firstly, our study included monitoring the Tc for a relatively short amount of time: 20 min of rest, ~10 min of exercise, and 40 min of recovery. The manufacturer of the CALERA Research sensor indicates that the device employs machine learning algorithms to determine the Tc. These algorithms may require more time for “training” to improve the prediction of the Tc. Therefore, future studies are warranted to test the validity of this device over longer periods of time. Another issue pertains to our sample size, which had enough statistical power but was relatively small and included only males, with only 14 enrolled patients from a single center. Furthermore, two patients had to be excluded as we were unable to collect data with the CALERA Research sensor due to excessive sweating. Overall, while we were able to collect enough data, enabling a robust comparison between temperature monitoring modalities, a larger sample size would allow for the stratification of our results into relevant clinical subgroups. Future studies should aim to assess the precision of the CALERA Research sensor among larger patient populations, stratifying the results by factors such as age, sex, and body mass index to obtain more comprehensive and generalizable findings.

## 5. Conclusions

Monitoring the Tc in individuals with CHF is crucial for their well-being and can help prevent complications. The availability of non-invasive sensors like the CALERA Research sensor offers promising solutions for monitoring the Tc, especially during effort, and can be used as a useful tool to protect vulnerable individuals from the adverse effects of heat exposure. However, the present study found that the Tc predicted by the CALERA Research sensor is poorly associated with the Tc assessed via gastrointestinal capsules during rest, exercise, and recovery. Also, the core temperature values obtained by the two methods were statistically significantly different, while the limits of agreement were relatively wide. These findings suggest that the CALERA sensor does not provide an accurate Tc estimation in CHF patients at rest and during brief exercise cardiopulmonary testing, as well as during recovery. Despite its limitations in providing accurate Tc estimations for CHF patients at various stages, the CALERA sensor stands out as a multifaceted monitoring device. Its ability to capture data on parameters such as skin temperature and heart rate provides multiple research opportunities. If its validity is improved, this device provides a budget-friendly alternative method, making it a viable option for extensive studies or those with budgetary restrictions.

## Figures and Tables

**Figure 1 sensors-24-00807-f001:**
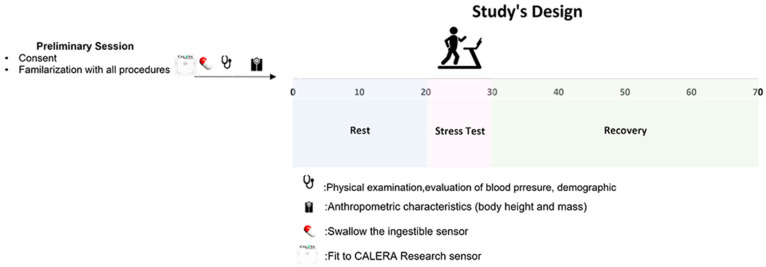
The study protocol.

**Figure 2 sensors-24-00807-f002:**
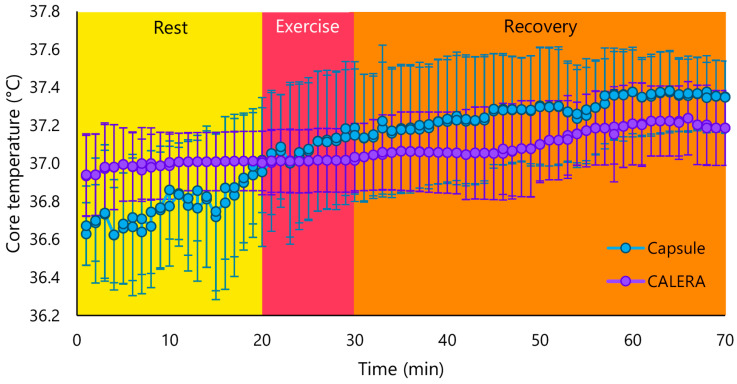
The Tc of the participants measured with the gastrointestinal sensor (blue line) and the CALERA Research sensor (purple line).

**Figure 3 sensors-24-00807-f003:**
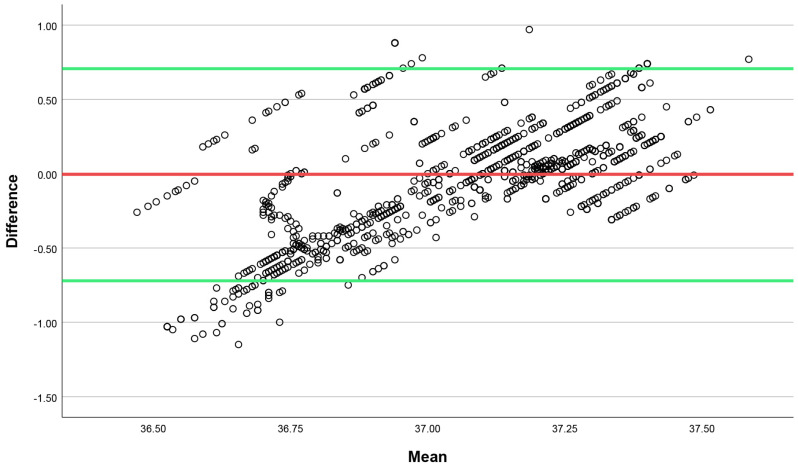
Bland–Altman plot of the differences between the Tc data from the gastrointestinal pill and the CALERA Research sensor. The red line indicates the mean difference, while the green lines indicate the upper and lower limits of agreement.

**Table 1 sensors-24-00807-t001:** Characteristics of CHF patients (mean ± SD).

**Age (years)**	53.25 ± 8.5
**Sex (male/female)**	(8/4)
**Height (cm)**	174.16 ± 2.7
**Weight (kg)**	95.5 ± 7.4
**BMI**	31.4 ± 2.4
**Left ventricle ejection fraction (%)**	35.6 ± 5.0
**Years of disease**	4.3 ± 1.6
**Etiology**	
**Ischemic**	66.7%
**Idiopathic dilated cardiomyopathy**	25%
**Hypertensive**	8.3%
**NYHA functional class**	
**I**	25%
**II**	58.3%
**III**	16.7%
**Current medications**	
**ACE inhibitors**	100%
**Diuretics**	66.7%
**Β-adrenergic receptor blocker (carvedilol)**	100%
**Spirolactone**	16.7%

Key: BMI = body mass index.

**Table 2 sensors-24-00807-t002:** Mean core body temperatures measured by the gastrointestinal pill and the CALERA Research sensor during periods of rest, exercise, and recovery.

	Gastrointestinal Pill (°C)	CALERA Research (°C)	*p*	Difference Mean (°C) [95% CI]	Cohen’s d	Pearson’s r
**Rest**	36.88 ± 0.37	37.00 ± 0.17	<0.001	−0.14 ± 0.80 [−0.94, 0.66]	0.40	−0.059
**Cardiopulmonary testing**	37.09 ± 0.35	37.04 ± 0.15	0.026	0.06 ± 0.38 [0.01, 0.11]	0.38	−0.015
**Recovery**	37.27 ± 0.28	37.16 ± 0.18	<0.001	0.11 ± 0.28 [0.09, 0.13]	0.28	0.292

Note: Confidence interval (CI). Significantly different mean temperature between the gastrointestinal pill and the Calera Research sensor when *p* < 0.05.

## Data Availability

Data are contained within the article.

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
