# Peer review of "Validity of the CALERA Research Sensor to Assess Body Core Temperature during Maximum Exercise in Patients with Heart Failure"

_sensors, 2024, doi:10.3390/s24030807_

Round 1

Reviewer 1 Report (Previous Reviewer 3)

Comments and Suggestions for Authors

The main ideas of the studies are better outlined in this version. 

A final question: does the study indicate whether a less hot environment (less than 28°C , as stated in line 304) can be beneficent for CHF patients?

Comments on the Quality of English Language

English language has been revised; the ideas can be more easily understood

Author Response

Reviewer 1

The main ideas of the studies are better outlined in this version.

Comment #1: A final question: does the study indicate whether a less hot environment (less than 28°C, as stated in line 304) can be beneficent for CHF patients?

Response #1: Thank you for raising this important question. Indeed, the choice of environmental temperature is a crucial factor for the safety and well-being of patients with chronic heart failure (CHF). Our study indicates that a temperate environment, specifically one with temperatures below 28°C, can be more beneficial for CHF patients. However, it's important to note that the overall safety and effectiveness of exercise for these patients are influenced not only by the ambient temperature but also significantly by their fitness level and the medications they are on, especially beta-blockers.

Reviewer 2 Report (Previous Reviewer 1)

Comments and Suggestions for Authors

The manuscript is acceptable after revision. 

Author Response

Reviewer 2

Comment #1: The manuscript is acceptable after revision.

Response #1: We appreciate your kind feedback.

Reviewer 3 Report (New Reviewer)

Comments and Suggestions for Authors

In this paper, the author proved the validity of the GALERA Research sensor when monitoring patients with CHF during periods of rest, throughout brief cardiopulmonary exercise testing, and during their subsequent recovery by comparing the monitoring results of gastrointestinal capsules and the non-invasive sensor called CALERA Research based on the body core temperature of individuals with chronic heart failure during rest or exercise of 12 patients. This article has a complete structure and broad application prospects, but there are still numerous problems in the article, and the following are the revisions:

(1) In page 1 line 29, the authors mentioned that "The CALERA 28 sensor is a practical, promising and cost-effective device… ". The discussion in this paper does not involve the manufacturing cost and economic benefits of the sensor, so it is considered to add the necessary expression to achieve logical self-consistency.

(2) In page 1 line 34, the authors mentioned that "Climate change and global warming pose a significant threat to human health.". The references to climate change and global warming here are not very relevant to the theme of this paper, and the reasons for people's concern are more far-fetched, so it is suggested to consider deleting them.

(3) In page 4 line 143, Figure 1 drawn by the author cannot clearly reflect the process and content of the experimental scheme, and the drawing is rather sloppy. It is suggested to modify the format of the chart.

(4) In page 3 line 138 and page 4 line 174, the author does not explain the ECG abbreviation, but adds a detailed description of the abbreviation.

(5) In page 7 line 237, in Figure 2, the monitoring results of gastrointestinal capsules are very different from those of the GALERA Research sensor. At the beginning of the exercise, it became clear that there were gastrointestinal capsules clear turning point in the monitoring results of gastrointestinal capsules. After the exercise, there was also an obvious fluctuation in the monitoring results of gastrointestinal capsules. However, the monitoring result of the GALERA Research sensor has been stable without any change. This does not lead to the conclusion that the error of the two test results is small, and it is suggested to add appropriate expressions.

(6) In page 7 line 237, in Figure 2, the curve color of A is not green. It is suggested to replace it with other colors to avoid misunderstanding.

In summary, although CALERA Research mentioned in this paper has great application prospects, the data processing is relatively rough, the number of subjects is small, and all are male, the conclusion is a little hasty and the reasons are not sufficient.

Author Response

Reviewer 3

General Comments

In this paper, the author proved the validity of the GALERA Research sensor when monitoring patients with CHF during periods of rest, throughout brief cardiopulmonary exercise testing, and during their subsequent recovery by comparing the monitoring results of gastrointestinal capsules and the non-invasive sensor called CALERA Research based on the body core temperature of individuals with chronic heart failure during rest or exercise of 12 patients. This article has a complete structure and broad application prospects, but there are still numerous problems in the article, and the following are the revisions:

Comment #1: In page 1 line 29, the authors mentioned that "The CALERA 28 sensor is a practical, promising and cost-effective device… ". The discussion in this paper does not involve the manufacturing cost and economic benefits of the sensor, so it is considered to add the necessary expression to achieve logical self-consistency

Response #1: Thank you for your comment. We have removed the term 'cost-effective' as per your suggestion.

Comment #2: In page 1 line 34, the authors mentioned that "Climate change and global warming pose a significant threat to human health.". The references to climate change and global warming here are not very relevant to the theme of this paper, and the reasons for people's concern are more far-fetched, so it is suggested to consider deleting them.

Response #2: Thank you for your insightful comment. We appreciate your concern regarding the direct relevance of the mention of climate change and global warming in our paper.  We have rephrased the sentence into “Global warming and the resulting climate change present a major challenge to public health”.

Comment #3: In page 4 line 143, Figure 1 drawn by the author cannot clearly reflect the process and content of the experimental scheme, and the drawing is rather sloppy. It is suggested to modify the format of the chart.

Response #3: A graphic designer has made corrections to the image as suggested.

Comment #4: In page 3 line 138 and page 4 line 174, the author does not explain the ECG abbreviation, but adds a detailed description of the abbreviation.

Response #4: Thank you for your comment. We have clarified that the abbreviation of ECG is electrocardiogram.

Comment #5: In page 7 line 237, in Figure 2, the monitoring results of gastrointestinal capsules are very different from those of the GALERA Research sensor. At the beginning of the exercise, it became clear that there were gastrointestinal capsules clear turning point in the monitoring results of gastrointestinal capsules. After the exercise, there was also an obvious fluctuation in the monitoring results of gastrointestinal capsules. However, the monitoring result of the GALERA Research sensor has been stable without any change. This does not lead to the conclusion that the error of the two test results is small, and it is suggested to add appropriate expressions.

Response #5: Thank you for highlighting the need for further explanation regarding the differences observed in Figure 2 between the gastrointestinal capsules and the CALERA Research sensor. Upon closer examination of the figure, it is noted that the temperature variation registered by the CALERA Research sensor is approximately 0.3°C. Similarly, the comparative difference between the two methods (gastrointestinal capsule and CALERA sensor) is around 0.3°C across all stages of the experiment.

To address your point about the interpretation of these results, we stated in the Discussion Section (page 8, lines 267-271), “Τhe comparative difference between the two methods (gastrointestinal capsule and CALERA sensor) is around 0.3°C across all stages of the experiment. The results imply that while the CALERA sensor offers a convenient method for estimating core temperature, its accuracy in providing precise Tc readings in CHF patients, both at rest and during exercise testing, might be limited.” This statement is intended to clarify that despite the small numerical difference in temperature readings, the CALERA sensor's consistency and lack of fluctuation, in contrast to the gastrointestinal capsules, indicate a potential discrepancy in accuracy for Tc estimation under the specific conditions tested.

Comment #6: In page 7 line 237, in Figure 2, the curve color of A is not green. It is suggested to replace it with other colors to avoid misunderstanding.

Response #6: Thank you very much for your comment, we have corrected it.

Comment #7: In summary, although CALERA Research mentioned in this paper has great application prospects, the data processing is relatively rough, the number of subjects is small, and all are male, the conclusion is a little hasty and the reasons are not sufficient.

Response #7: Thank you very much for sharing with us your concerns. In our limitations paragraph we mentioned all these. Page 9, lines 318-332

“Despite the valuable insights gained from our study, several limitations should be acknowledged. Firstly, our study included monitoring Tc for a relatively short amount of time: 20 min of rest, ~10 min of exercise, and 40 min of recovery. The manufacturer of the CALERA Research sensor indicates that the device employs machine learning algorithms to determine Tc. These algorithms may require more time for "training" to improve the prediction of Tc. Therefore, future studies are warranted to test the validity of this device over longer periods of time. Another issue pertains to our sample size, which had enough statistical power but was relatively small and included only males, with only 14 enrolled patients from a single center. Furthermore, two patients had to be excluded as we were unable to collect data with the CALERA Research sensor due to excessive sweating. Overall, while we were able to collect enough data enabling a robust comparison between temperature monitoring modalities, a larger sample size would allow stratifying our results into relevant clinical subgroups. Future studies should aim to assess the precision of the CALERA Research sensor among larger patient populations, stratifying the results by factors such as age, sex, and body mass index to obtain more comprehensive and generalizable findings.”

Round 2

Reviewer 3 Report (New Reviewer)

Comments and Suggestions for Authors

All is Ok. It can be accepted.

This manuscript is a resubmission of an earlier submission. The following is a list of the peer review reports and author responses from that submission.

Round 1

Reviewer 1 Report

Comments and Suggestions for Authors

This manuscript reports an optimized method to control the yarn tension through a cascade control of tension and position with feedback controllers. The paper is completed in the present form, but it may attract limited interest for the readers of this journal. The reviewer recommends publishing it in the journal. The reviewer recommends publishing it in the journal. 

Reviewer 2 Report

Comments and Suggestions for Authors

This paper presents the validity of a non-invasive sensor for measuring core body temperature. I think validation studies are important, so I recommend improving the paper so it can be published.

Below are some points that should be improved.

TITLE: Add the name of the sensor

INTRODUCTION

Lines 53-58: You talk about working out in hot environments. To my knowledge, core temperatures when working out in hot environments (above 38 or 39 °C) are significantly higher than those reported in your study.

Line 75: It might be worth adding the information that the same sensor used to be called CORE.

Line 79: Improve the last sentence.

MATERIALS AND METHODS:

Lines 83-95: Data of the participants is missing (mass, height, age,...etc.)

Lines 97 - 115: Describe in more detail what the first and second visit was.

Lines 111-115: I think this should be part of 2.3.

Line 124: Describe what the Bruce protocol is.

Lines 147-162: This is not important for the validation study.

Lines 163-175: I think this part should be under statistical analyses

Line 184: I don't think you mentioned before that it was e-Celsius.

RESULTS

Line 196: This should be part of the methods.

Figure 2. The results shown in Figure 2 are quite strange. There is a visible increase in temperature during rest (by about 0.6°C), while the increase during exercise is minimal, and there is also minimal increase during recovery (measured with the pill). Can you explain these results? To my knowledge, I would expect the temperature to remain constant during rest, increase during exercise, increase some more at the beginning of the recovery phase, and then decrease again. Most importantly, please explain why the temperature rises during the rest phase. Otherwise, you might think that you didn't wait long enough after taking the pill, which would make the study irrelevant.

Reviewer 3 Report

Comments and Suggestions for Authors

The paper presents a sensor, called GALERA sensor, which is used to monitor the core body temperature during exercises for patients with chronic heart failure.

The title is somewhat confusing, please reformulate to express more clearly the main idea of the paper.

Reformulate the sentence for a better understanding: "This study aimed to assess the validity of the GALERA Research sensor at rest, during brief exercise cardiopulmonary testing, and recovery in patients with CHF" (line 16-17) - the patients are at rest or doing exercises?

"The Tc values from the CALERA research sensor and the gastrointestinal sensor showed no associations at rest (r=0.056, p=0.154) and during exercise (r=-0.015, p=0.829) and a weak association during recovery (r=0.292, p<0.001)." : the associations should have been between ... ?

"Conclusions: The CALERA sensor is a practical, promising and cost-effective device, but it does not provide an accurate Tc estimation in CHF patients at rest, during brief exercise testing, and during recovery (line 27-28)": which are the advantages of using the CALERA  sensor then?

Are there statistical data that more heart failures occur in warmer seasons than in cold ones? 

Please explain more detailed how the Calera sensor works. Is it an invasive or a non-invasive procedure? "The Tc was recorded continuously using a telemetric capsule" - the procedure is still invasive if there is need of the ingestible capsule.

Which is the functional relation between the studied CALERA sensor and the mentioned CORE sensor?

Conclusions are very poor. Please detail the significance of the results.

Which are the limitations of the study?

Comments on the Quality of English Language

The English Language should be revised by a native speaker

Round 2

Reviewer 2 Report

Comments and Suggestions for Authors

The authors have improved the manuscript, but with additional explanations I find some problems in their methodology.

As stated in https://journals.lww.com/acsm-msse/fulltext/2008/03000/the_effect_of_cool_water_ingestion_on.18.aspx, using the GI pill is a valid measure of T_rec for about 30-60 minutes after ingesting cold fluids. From their explanation, I understand that the participants drank the water immediately before taking the pills and only 10 minutes were subtracted from the measurement, which is insufficient. Therefore, the results are not ''correct'' and cannot be used for a validation study.

I see an even bigger problem in the fact that they only wait 10 minutes after taking the pill. It is common knowledge that you should wait longer after taking the pill and this has been published in several publications (at least 40 minutes to my knowledge). Therefore, I believe that these results should not be published as the methodology has some major flaws.

Author Response

Point-to-Point response to the Reviewer's comments

REVIEWER #2

GENERAL COMMENT: The authors have improved the manuscript, but with additional explanations I find some problems in their methodology.

RESPONSE #1: Thank you for your positive comment.

SPECIFIC COMMENTS:

COMMENT #1: As stated in https://journals.lww.com/acsm-msse/fulltext/2008/03000/the_effect_of_cool_water_ingestion_on.18.aspx, using the GI pill is a valid measure of T_rec for about 30-60 minutes after ingesting cold fluids. From their explanation, I understand that the participants drank the water immediately before taking the pills and only 10 minutes were subtracted from the measurement, which is insufficient. Therefore, the results are not ''correct'' and cannot be used for a validation study.

RESPONSE #1: Thank you for your insightful comment and for highlighting the potential impact of cold water ingestion on core temperature measurements using a gastrointestinal (GI) pill. We agree with the results of Wilkinson et al. 2008, which showed that ingesting cold fluids can affect T_rec readings for approximately 30-60 minutes. In this study, participants consumed 250ml of cold water prior to the intervention. However, there is a misunderstanding, since in our study, patients were asked to swallow the capsule and drink cold water to ensure hydration. As mentioned on page 3 and lines 117 and 118, the time between swallowing the capsule and the onset of the protocol was 121   ±  17 min.  This duration was chosen to ensure accurate temperature readings post-ingestion of cold water, especially considering the specific needs of our participants, who were patients with heart failure. This aligns with the methodology used in the study by Notley et al. (2021), which suggests that the timing of pill ingestion does not significantly influence the validity of telemetry pill temperature measurements, as an index of core temperature.

COMMENT #2: I see an even bigger problem in the fact that they only wait 10 minutes after taking the pill. It is common knowledge that you should wait longer after taking the pill and this has been published in several publications (at least 40 minutes to my knowledge). Therefore, I believe that these results should not be published as the methodology has some major flaws.

RESPONSE #2: Thank you for raising this concern about the waiting period after ingesting the telemetry pill. We agree with you that a 10 min waiting period after taking the capsule is a short period is a short period. However, it seems there has been a misunderstanding regarding the methodology described in our manuscript.

In our manuscript, as detailed on Page 3 (Lines 120-121), we explicitly state, "The time between swallowing the capsule and the onset of measurements was 121 ± 17 minutes." This information is reiterated for clarity on the same page (Lines 130-131) in the revised version. This waiting period of approximately 2 hours (121 minutes) was intentionally chosen to align with the best practices noted in the literature, which typically suggests a waiting period of at least 40 minutes to ensure accurate core temperature readings from telemetry pills. The mention of a 10-minute period in our study refers to a different aspect of the methodology, not related to the initial waiting time post-pill intake. This might have led to the confusion. The 121-minute waiting period was consistently adhered to across all participants to ensure that the influence of external factors, such as the ingestion of cold water or the initial stabilization of the telemetry pill, was minimized. This approach was taken to ensure the reliability and validity of our core temperature measurements. We believe this clarification addresses the concerns regarding the methodology and reaffirms the integrity of our study's findings.
